# Research on Meshing Gears CIMT Design and Anti-Thermoelastic Scuffing Load-Bearing Characteristics

**DOI:** 10.3390/ma15062075

**Published:** 2022-03-11

**Authors:** Xigui Wang, Jian Zhang, Yongmei Wang, Chen Li, Jiafu Ruan, Siyuan An

**Affiliations:** 1School of Engineering Technology, Northeast Forestry University, No. 26, Hexing Road, Xiangfang District, Harbin 150040, China; lcnefu@163.com (C.L.); rjfnefu1993@126.com (J.R.); asy1996@126.com (S.A.); 2School of Mechatronics and Automation, Huaqiao University, No. 668 Jimei Avenue, Jimei District, Xiamen 361021, China; 3School of Mechanical and Electrical Engineering, Northeast Forestry University, No. 26, Hexing Road, Xiangfang District, Harbin 150040, China; zjnefu1980@126.com; 4School of Motorcar Engineering, Heilongjiang Institute of Technology, No. 999, Hongqidajie Road, Daowai District, Harbin 150036, China

**Keywords:** meshing gears, contact interface micro-texture, thermoelastic hydrodynamic lubrication, fractal microstructure design, characteristic analysis

## Abstract

In the process of gear meshing, it is an inevitable trend to encounter failure cases such as contact friction thermal behavior and interface thermoelastic scuffing wear. As one of the cores influencing factors, the gear meshing contact interface micro-texture (CIMT) significantly restricts the gear transmission system (GTS) dynamic characteristics. This subject suggests the contact characteristic model and interface friction dynamics coupling model of meshing gear pair with different CIMT. Considering the influence of gear meshing CIMT on distribution type of hydrodynamic lubricating oil film, contact viscous damping and frictional thermal load, the aforementioned models have involved transient meshing stiffness (TMS) and static transmission accumulated error (STAE). Based on the proposed models, an example verification of meshed gear pair (MGP) is analyzed to reveal the influence of CIMT on the dynamic characteristics of GTS under a variety of micro-texture configurations and input branch power and rated speed/shaft torque conditions. Numerical simulation results indicate that the influence of CIMT on gear dynamic response is extremely restricted by the transient contact regularity of the meshing gear surface. Meshing gears’ dynamic characteristics (especially vibration and noise) can be obviously and effectively adjusted by setting a regular MGP with CIMT instead of random gear surfaces.

## 1. Introduction

Involute cylindrical gears are often used in power rear transmission systems to satisfy the requirements for precise delivery index of motion, while having high power density and low friction loss. These cylindrical gears are often subjected to impact-type loads on them, which can naturally cause periodic oscillation due to the instantaneous MGP. The GTSs have been extensively focused on many cores’ industrial categories such as naval architecture, aerospace and ocean engineering, and wind power generation (roadbed and seabed).

The MGP contact interface with low speed and light load may have a thicker dynamic pressure oil film, and the meshing friction coefficient is small. The oil film pressure and temperature of the MGP interface with heavy-duty are relatively high; the viscosity-temperature effect of lubricating oil rapidly reduces the thickness of the interface oil film, which further leads to pitting corrosion and even thermal scuffing or wear on the meshing interface. The friction characteristics of the bearing seat of the recessed bushing manufactured by the processing technology and chemical etching technology have been studied [1]. The load, oil varieties, groove size, depth and shape are analyzed through experiments to examine the lubricating effect of the pit texture to explore their influence on the friction characteristics. The research results explained that the friction performance of the journal bearing is improved by designing the size of the pit texture reasonably. The sleeve has better friction properties. Bushings with etched pits on the entire circumference have better friction performance than bushings with grooves engraved on half of the circumference. Especially in hybrid thermal elastohydrodynamic lubrication (TEHL) systems, the secondary lubrication effect produced by pits is the main mechanism to improve performance.

Based on this, how to improve the dynamic pressure lubrication effect of the meshing tooth surface (MTS) by changing the MGP with CIMT has become one of the core directions in the research field of gears [2]. With the gradual development of laser technology, the micro-texture process performed by a laser can significantly improve the TEHL performance of the contact surface [3]. After laser treatment, the textured MTS is customized with micron-level dynamic pressure lubrication structure grooves or pits that can store lubricating oil, which can significantly improve the compressive resistance of the MTS, thereby increasing the anti-scuffing load-carrying capacity of the MGP and providing pre-research solutions for performance and response to extreme working conditions.

At present, the research of interface micro-texture mainly focuses on low-pair contact, focusing on the geometric shape design of micro-texture and the selection and setting of its size parameters. Researchers mostly recognize these surface micro-texture design shapes, such as circular pits, grooves, and convex hulls [4]. The feasibility study of theory and experiment on micro-pits using laser surface texture technology was performed. A regular surface texture with a micro-pit shape is formed on the contact interface of the friction machine, and the bearing capacity, wear resistance, friction coefficient and other aspects of the friction machine components have been significantly improved [5,6]. The research results show that, in the case of full lubrication or hybrid TEHL, each micro-pit can be used as a miniature hydrodynamic lubrication bearing or as a miniature storage device for lubricant. The circular pits distributed at intervals are processed by laser on the surface of the piston ring. The influence of the diameter, spacing and distribution position of the pits on the friction coefficient is analyzed, and the relevant influence parameters are optimized and analyzed, based on the joint solution of the Reynolds equation and the radial motion equation. They then reveal the fluid dynamic pressure distribution law between the cylinder inner liner and piston ring [7]. With the aid of the material ablation process and the use of pulsating laser beams to create thousands of micro-pits arranged vertically and horizontally on the metal surface, tribological experiments are carried out in the range of 0.015–0.75 m/s through the disc needle device. The test results show that the laser-textured surface under the action of hydrodynamic lubrication can greatly reduce the friction coefficient, and that the lower pit density is more conducive to the lubrication of the metal surface. In the case of higher moving speed, load and higher viscosity oil, the effect of micro-texture on the metal surface is more obvious [8]. Based on a numerical simulation method of the virtual texture model, the influence of the bottom shape of the texture and the relative movement of the surface on the generated texture is studied. These textures are located on a triangularly distributed interaction surface and have the same density. The simulation results show that there is a thicker lubricating oil film related to the bottom configuration of the micro-element wedge and the micro-step hydrodynamic bearing [9]. The regularly distributed circular micro-textures are processed by laser on the stainless-steel surface, and the friction-reducing mechanism of micro-textures is studied with the numerical simulation method, which in turn reveals the law of oil film pressure change in the range of circular pit diameter from 20.0–1000 μm, pit depth within 1.0–100 μm and pit density within the range of 3–62% [10]. The analysis results show that the depth of the micro-texture is determined by the load. As the load increases, the depth of the micro-texture pits increases accordingly to ensure a small contact interface friction force. The friction force of the micro-pit surface based on hydrodynamic lubrication is reduced by nearly 80% compared with the untextured surface.

Although the above-mentioned scholars experimentally verified the dynamic pressure lubrication effect of the micro-texture of the contact interface processed by laser and obtained the specific optimal micro-texture pit size, the basis for the selection of the micro-texture geometry and the optimization theory are not described in detail. The optimization form is mostly enumerated by experiments, facing many micro-textured geometric figures and their distribution patterns, which will make it difficult to obtain the most effective dynamic pressure lubrication prediction scheme. Based on this, some scholars have tried to establish a parameter model that optimizes the relationship between the micro-texture geometry and the friction performance, and that develops the micro-texture pit geometry, depth-to-diameter ratio, zone occupancy, local texture and self-lubricating material filling and other aspects [11,12,13]. The micro-textured surface has a crucial correlation effect on the contact tribological performance, especially the micro-texture shape and micro-texture bottom profile; micro-texture direction and micro-texture density are the most important key parameters affecting oil film thickness and friction coefficient [14]. The micro-textured triangle effect is the most obvious in reducing friction [15]. Based on the combination of numerical analysis and experiment, the size and distribution of micro-texture can be optimized to a certain extent, and then the micro-texture contact interface structure can be innovatively designed. It is well known that the micro-texture design method and design criteria are not unified, and that the experimental data and its conclusions are quite different. In addition, the experimental process and optimization analysis are too cumbersome. In view of the diversity and infinity of micro-texture shapes, it is impossible to determine which interface micro-texture form is most beneficial to reduce contact friction and wear.

How to reduce the cases of micro-texture forms as much as possible and find the most superior micro-texture shape becomes the most critical issue. Many friction contact interfaces have been successfully micro-textured, such as turning tools, cylinder inner liners, dynamic and static sealing rings, MGPs, rolling/sliding bearings, oil-gas mixing/separation equipment, silicon-controlled wafer surface, and micro-nanomechatronics systems of various intelligent machine tools. Summarized in Figure 1, the micro-texture of the contact interface shows its unique personality characteristics [16,17,18,19]. A pit-type micro-textured surface, based on bionics, that is anti-friction and resistance-reducing, self-cleaning, and wear-resistant was designed. The bionic micro-textured surface is shown in Figure 2 [20,21,22].

With the improvement of high-speed and heavy-duty GTS performance indicators, the dynamic responses of GTS have been extremely valued in the preliminary design and verification and optimization stages of MGPs. The transient contact characteristics of the MGP interfaces are one of the core related influencing indexes that restrict the GTS dynamic response. In particular, the CIMT is an extremely interesting feature of the MGP surfaces. Various types of CIMT may lead to various dynamic responses of the GTS.

Many monographs are related to the dynamic characteristics of the GTS [23,24,25,26]. The TMS excitation injects a time-varying meshing parameters into the GTS dynamic analysis equation as a key indicator, which are of great concern; it has constituted the intrinsic properties of the dynamics of MGPs, which, in turn, forms the characteristics and inherent properties of the GTS dynamics [27,28,29]. The numerical analytical methods for solving the TMS of MGPs are divided into two categories: the finite element method (FEM) [30] and the analytical method following the principle of potential energy [31]. The potential energy method occupies very little computational resources compared with the FEM, but it can still achieve extremely accurate numerical results comparable to the FEM analysis data. Some scholars have considered various stimulating factors, such as TMS, MTS defects, and optimization analysis of gear tooth profile modification, etc., and have carried out research on the dynamic characteristics of related GTS [32,33,34]. However, most researchers in academia are more inclined to study the CIMT when studying the interaction of MGP contact interface lubrication enrichment effect on the GTS dynamic load-bearing characteristics, whereas the CIMT is abandoned on most levels. This can easily lead to misunderstandings about the effect of the surface contact micro kurtosis of meshing tooth as the CIMT can also produce unexpected results in the gear dynamics. Herein, considering the appearance effect of CIMT’s micro-element body, and then eliminating the adverse influence of MTS contact micro kurtosis, this work intends to study the dynamic characteristics of gears with the same MTS contact micro kurtosis but different CIMT. This is very likely to provide new ideas for improving the anti-thermoelastic scuffing load-bearing characteristics of GTS by controlling the micro-texture of the contact interface. Based on the fractal theory, the M-B rough surface elastoplastic contact fractal model is used to study the microscopic surface contact characteristics, the relationship between the contact zone of the asperities and the normal load is determined, and the fractal dimension is used to characterize the complexity of different surface profile features and irregularity, which describes the geometric characteristics of sections and curved surfaces with fractal geometry. This subject intends to use fractal graphics as the basis for the design of the micro-texture of the contact interface and to analyze the changing trend between the fractal dimension and friction and wear of different fractal graphics, so as to obtain the best fractal micro-texture, which is the micro-texture of the MTS design, as well as to improve the contact interface lubrication and drag reduction to provide new technology accumulation.

## 2. Gear Dynamic Model with CIMTs

In this sub-project, a gear friction dynamics model with CIMT is proposed, which mainly includes gear dynamics models with different CIMT evolutions and gear 3D TEHL transient meshing models to explore the correlation effect of CIMT on GTS dynamic characteristics. The GTS friction dynamics coupling equations with CIMTs are solved by an iterative loop between the above two models, as shown in Figure 3.

In the absence of iterative loop solution, first initialize the basic parameters of the MGP, such as the comprehensive error e(t) of gear tooth manufacturing, rated input torque and its corresponding speed. The TMS $k_{m}(t)$ is analyzed according to the potential energy method. If it is not iterated, the starting value for viscous damping and sliding/rolling friction are set to $F_{s}=0,C_{m}=0$. Considering the mentioned above excitations, the vibration equation of the MGP dynamics model can be solved according to the Runge–Kutta analytical method. The sliding speed $u_{1}(t),u_{2}(t)$ and curvature radius R(t) of the tooth surface at the meshing position are predicted by the GTS vibration motion analysis. At the same time, the dynamic meshing force (DMF) $F_{T}(t)$ is deduced by using the vibration transient response of the GTS, and then substituting it into the interface impact preset load model (distributed form) to derive the DMF $F_{T}(t)$ of the MGP. For the instantaneous values of $u_{1}(t),u_{2}(t)$, R(t), $F_{T}(t)$, as well as the CIMT and lubrication enrichment effect characteristics, a typical semi-system method is used to analyze the control equations of the hybrid TEHL model [35]. The rolling friction $F_{r}$, the sliding friction $F_{s}$ and the viscous damping $C_{m}$ of the gear meshing interface are subject to the fast-converging numerical solution of the effective model of the hybrid TEHL enrichment effect, which is used to calculate the updated real-time follow-up response of the MGP dynamic model proposed above.

An iterative cycle process is always executed until the DMF of the MGP must meet the convergence criterion. The criterion for convergence of the iterative loop is as follows:(1)$$\frac{\sum_{\gamma }\left|F_{p}^{k}(t_{\gamma })-F_{p}^{k-1}(t_{\gamma })\right|}{\sum_{\gamma }F_{p}^{k}(t_{\gamma })}<err$$
where *γ* represents the tooth pair mesh position in a meshing period, *k* the number of steps in the loop iteration, $F_{p}^{k}(t_{\gamma })$ denotes the calculated DMF at $t_{\gamma }$ (instantaneous meshing time) calculated in the K-th loop iteration procedure, and the greater influence parameter err is a pre-defined threshold for fast convergence, whose predetermined threshold is set to $10^{-5}$ in the research issue. Thus, it can be deduced, whenever the relative error value between the DMFs analyzed in two consecutive procedures is not greater than the pre-defined value, that, once the iterative loop process stops, it is assumed that a stable solution is determined.

In the actual operation of GTS, the micro-appearance morphologies of MTS directly depend on the processing accuracy, manufacturing process, operation law and constituent materials, etc. The related discussion focuses on the three different MTS micro-appearance morphologies that exist in actual processing and manufacturing. It is set in advance that these three designed cases all have the same MTS contact micro kurtosis R_q_ (root mean square (RMS) contact micro kurtosis value of MTS) and the same wavelength L; however, the cross-sections of the two are not equal, and are marked by random permutation (Case 1), sinusoidal distribution (Case 2), and semi-ellipse designed configuration (Case 3), respectively, as outlined in Figure 4. The associated influence of CIMT on the friction characteristics and dynamic responses of the MTS is further revealed.

Figure 5 depicts a general spur gear dynamic model considering MTS friction. Herein, the representative parameters $F_{fp}$ and $F_{fg}$, which describe separately the friction behavior of pinion (driving gear) and bull gear (driven gear). The transient meshing description of a spur meshing pair is modeled simultaneously by TMS $k_{m}(t)$ and viscous damping $C_{n}$. The support bearing of each gear is set to real-time simulation of equivalent support stiffness and interface contact equivalent damping indicated by the two specified directions of *x* and *y*; the relevant parameters are $k_{px}$, $k_{gx}$, $k_{py}$, $k_{gy}$ and $C_{px}$, $C_{gx}$, $C_{py}$, $C_{gy}$. The driving/driven gears in the dynamic models are represented by the equivalent simplified representation of a rigid body whose mass $m_{i}(i=p,g)$, moment of inertia $J_{i}$ and radius are equivalent to the radius $r_{i}$ of the base circle of the preset gear. Assume that the rotational and translational movements of the two gears in the y direction are coupled along the meshing interface contact line of action (LOA) by the spring damping unit. $k_{m}$ represents the TMS of the MGP, and $C_{n}$ describes the equivalent meshing interface damping of the MGP. e is regarded as the STAE, which mainly includes meshing gear pair thermoelastic deformation and gear teeth manufacturing error under static load conditions. The rotational and translational degree of freedoms of the MGP in the x-direction tends to be coupled in real time in the off-line action (OLOA) direction. The control equations descriptions of the above gear dynamic model are expressed as:(2)$$J_{p}{\ddot{\theta }}_{p}=T_{p}-F_{p}(t)r_{p}-{\left[\sum_{i=1}^{n}F_{fp}(t)R_{p}(t)\right]}_{i}$$
(3)$$m_{p}{\ddot{y}}_{p}+c_{py}{\dot{y}}_{p}+k_{py}y_{p}=-F_{p}(t)$$
(4)$$m_{p}{\ddot{x}}_{p}+c_{px}{\dot{x}}_{p}+k_{px}x_{p}={\left[\sum_{i=1}^{n}F_{fp}(t)\right]}_{i}$$
(5)$$J_{g}{\ddot{\theta }}_{g}=T_{g}-F_{g}(t)r_{g}-{\left[\sum_{i=1}^{n}F_{fg}(t)R_{g}(t)\right]}_{i}$$
(6)$$m_{g}{\ddot{y}}_{g}+c_{gy}{\dot{y}}_{g}+k_{gy}y_{g}=F_{g}(t)$$
(7)$$m_{g}{\ddot{x}}_{g}+c_{gx}{\dot{x}}_{g}+k_{gx}x_{g}=-{\left[\sum_{i=1}^{n}F_{fg}(t)\right]}_{i}$$
where n represents the number of the MGPs, $T_{i}$ denotes the external loads acting on the MGPs i (i=p,g, is shown as the pinon and bull gear respectively), ${\dot{\theta }}_{i}$ and ${\ddot{\theta }}_{i}$ represent the torsional vibration velocity/acceleration (VVA) of the MGPs i, ${\dot{x}}_{i}$ and ${\ddot{x}}_{i}$ denote the translational VVA in the meshing interface OLOA orientation of the MGPs i, respectively, ${\dot{y}}_{i}$ and ${\ddot{y}}_{i}$ represent the translational VVA in the meshing interface LOA orientation of the MGPs i, respectively, $R_{i}$ represents the radius scalar at the meshing domain location of the MGPs i. The frictional forces $F_{fp}$ and $F_{fg}$ of the pinion/driven gears are described by the hybrid TEHL calculation model. The DMF $F_{p}$ is written as:(8)$$F_{p}=k_{m}(y_{p}+r_{p}\theta_{p}-y_{g}+r_{g}\theta_{g})+C_{n}({\dot{y}}_{p}+{\dot{r}}_{p}{\dot{\theta }}_{p}-{\dot{y}}_{g}+{\dot{r}}_{g}\theta_{g})-(k_{m}e+C_{n}\dot{e})$$

The tooth pair number in the meshing zone presents a periodic time-varying law during the meshing transient process of the spur MGP. Consider an MGP with a thermal steady state interfacial contact ratio between 1 and 2, the number of gear teeth in the alternate meshing. The entire meshing zone is set as a double teeth contact (DTC) zone and a single tooth contact (STC) zone. The overall meshing pair DMF is expressed as the sum of the DMF of all MGPs in the meshing zone:(9)$$F_{p}=F_{T1}+F_{T2}$$
where $F_{T1}$ and $F_{T2}$, respectively, denote the DMF of the first and second meshing gears teeth pairs, and $F_{T2}=0$ represents the DMF when only one tooth pair is in meshing. According to the load distribution model between meshing gear teeth, the DMF of each MGP is derived:(10)$$\left\{ \begin{array}{l} F_{T1}=k_{1}\delta_{1}+c_{1}{\dot{\delta }}_{1}\\ F_{T2}=k_{2}\delta_{2}+c_{2}{\dot{\delta }}_{2} \end{array}\right.$$
where $k_{i}$ is simulated by the TMS of the *i*-th MGP (i=1,2).$c_{i}$ represents the viscous damping index of the *i*-th MGP (i=1,2), in which is usually assumed to be an associated parameter dependent on $k_{i}$. The calculation formula can be expressed as:(11)$$c_{i}=\frac{k_{i}6(1-\alpha )}{V_{i}({\left(2\alpha -1\right)}^{2}+3)}$$
where $V_{i}$ represents the interface vibration velocity index when the gear pair is initially meshed relative to the parameter $\alpha =1-0.022V_{i}{}^{0.36}$. In the above Equation (11), δ and ${\dot{\delta }}_{i}$, respectively, denote the contact interface relative velocity and slippage displacement of the *i*-th MGP (i=1,2) along the LOA orientation, which can be indicated as follows:(12)$$\left\{ \begin{array}{l} \delta_{i}=(y_{p}-y_{g})+(r_{p}\theta_{p}+r_{g}\theta_{g})-e_{i}\\ {\dot{\delta }}_{i}=({\dot{y}}_{p}-{\dot{y}}_{g})+(r_{p}{\dot{\theta }}_{p}+r_{g}{\dot{\theta }}_{g})-{\dot{e}}_{i} \end{array}\right.$$
where $e_{i}$ indicates the displacement excitation caused by the meshing interface contact profile deviations (relative to the ideal standard involute gear profile) of the *i*-th MGP (where i=1,2) during an MGP meshing transient contact process, whereas ${\dot{e}}_{i}$ is shown as the derivative of the parameter $e_{i}$, which is a quantitative characterization of the velocity excitation.

## 3. Three-Dimensional TEHL Calculation Model of Gear MTS

Considering the comprehensive influence of factors such as MTS excitation load and slip speed, lubricating oil viscosity and tooth surface contact micro kurtosis, the lubrication conditions of MTS are divided into 3 situations: elastohydrodynamic, hybrid and boundary lubrications. Among the aforementioned factors, the MTS excitation load is calculated by Formula (11) to obtain the gear tooth DMF. The slip speed of the MGP is the transient difference between its linear speeds in the OLOA orientation, namely, $u_{p}(t)$ and $u_{g}(t)$, which is indicated as follows:(13)$$\left\{ \begin{array}{l} u_{p}(t)={\bar{u}}_{p}(t)-R_{p}(t){\dot{\theta }}_{p}(t)+{\dot{x}}_{p}(t)\\ u_{g}(t)={\bar{u}}_{g}(t)-R_{g}(t){\dot{\theta }}_{g}(t)+{\dot{x}}_{g}(t) \end{array}\right.$$
where $u_{i}(t)$ denotes the linear velocity associated with the nominal rotation of MGP i (i=p,g) in the orientation of OLOA of the meshing interface, and which is defined as $u_{i}(t)=-R_{p}(t)\omega_{i}$; herein, $\omega_{i}$ denotes the gear pair nominal rotational peripheral speed. $R_{p}(t){\dot{\theta }}_{p}(t)$ and $x_{i}(t)\;(where,\;i=p,g)$ respectively represent the velocity vector components in the orientation of OLOA of the meshing interface caused by rotation/translation vibration excitation.

Under hybrid TEHL conditions, the transient time-varying TEHL oil film and non-smooth contact interface in the gear meshing zone coexist. In the meshing zone where there is no concave-convex contact, the MGPs are secluded by a layer of hydrodynamic TEHL oil film, thereby forming the full oil film lubrication. The typical feature of gear meshing is usually in line contact with the fluid hydrodynamic lubricating oil between the gear teeth that participate in the meshing. Combined with the Reynolds equation, the expression of the commonly used 3D line contact TEHL model can be shown as:(14)$$\frac{\partial }{\partial x}\left(\frac{\rho }{12\eta^{\#}}h^{3}\frac{\partial p}{\partial x}\right)+\frac{\partial }{\partial y}\left(\frac{\rho }{12\eta^{\#}}h^{3}\frac{\partial p}{\partial y}\right)=\mu_{r}\left(\frac{\partial \rho h}{\partial x}+\frac{\partial \rho h}{\partial t}\frac{1}{\mu_{r}}\right)$$
where x is regarded as the relative slippage orientation and y is in the axial orientation of the MGP; p and h represent the characterization parameters of the lubricating oil between the contact interfaces: film thickness/pressure, respectively.

ρ denotes the lubricating oil density; the relative meshing interface rolling/slipping speed $\mu_{r}$ is shown as an average instantaneous speed of the MGP, which can be expressed as $\mu_{r}=1/2\left(\mu_{g}(t)+\mu_{p}(t)\right)$. $\eta^{\#}$ stands for equivalent kinematic viscosity index. In view of the non-Newtonian features of the lubricating oil shear thinning effect, the following expression can be obtained from the Ree–Eyring model:(15)$$\frac{1}{\eta^{\#}}=\frac{\tau_{0}}{\eta \tau_{m}}\sinh\left(\frac{\tau_{m}}{\tau_{0}}\right)$$
where $\tau_{0}$ is expressed as the reference value of the shear stress resistance of the lubricating oil, $\tau_{m}$ is shown as the arithmetic mean of the viscous shear stress resistance of the lubricating oil, η represents the viscosity of lubricating oil under low shear rate, which can be solved by Roelands’ formula.

The specific properties (rheological properties) for the hydrodynamic oil film at the contact interface are determined by the micro-texture morphology of the MTS. The hydrodynamic film is extremely thin at the peak abrupt position of the uneven MTS contact micro kurtosis, where the meshing gear tooth interface contact easily occurs. As a result, the original lubrication conditions are converted to hybrid TEHL. At a specific gear pair meshing position that, due to the microscopic morphology of the contact interface of the MTS, cannot be solved with the general form of the Reynolds equation, the thermoelastic contact theory needs to be sought.

In order to obtain a uniform numerical completeness for the contact zone of the entire meshing interface, once the hydrodynamic lubricating film thickness index is expressed as zero, all the kinetic related TEHL parameters items in the Reynolds equations are regarded as in a non-open state, which can satisfy the numerical solution of the initial contact pressure of the gear pair MTS. From the above, this can be equivalent to deriving the simplified form of the Reynolds equation as:(16)$$\mu_{r}\frac{\partial \rho h}{\partial x}+\frac{\partial \rho h}{\partial t}$$

In view of the fact that Equation (15) is regarded as another expression of Equation (16), for the numerical full solution of the TEHL problem, the interface pressure parameter automatically meets the non-intermittent conditions of the fluid dynamics boundary and the non-smooth contact zone, and no additional settings are required. Based on this, the description of hybrid TEHL is analyzed by the same repeated process of iterative looping until fast convergence. The mathematical form expression of the physical quantity of hydrodynamic oil film thickness in the TEHL zone of the meshing gear tooth contact interface can be expressed as:(17)$$h(x,y,t)=h_{0}(t)+\frac{x^{2}}{2R_{x}}+V(x,y,t)+\delta (x,y,t)$$
where $R_{x}$ represents the equivalent curvature radius of the MGPs at meshing point, which is set to $\frac{1}{R_{x}}=\frac{1}{R_{p}}+\frac{1}{R_{g}}$. *δ* indicates the geometric shape distribution parameter of the composite micro-topography of the two pairs of teeth surfaces at the meshing point of the driving and driven gears, which are summarized in Figure 6. *V* is regarded as the thermal elastic deformation parameter of the MTS and can be analyzed by the DC-FFT method. Then its numerical calculation expression is:(18)$$V(x,y,t)=\frac{2}{\pi E^{\prime }}\iint_{\Omega }\frac{p_{f}(\xi ,s)+p_{c}(\xi ,s)}{\sqrt{{\left(x-\xi \right)}^{2}+{\left(y-s\right)}^{2}}}d\xi ds$$
where $p_{f}$ denotes the film pressure of hydrodynamic oil, which can be achieved by analytic coupling Equations (14), (17) and (18). $p_{c}$ represents the non-smooth interface contact micro kurtosis pressure, which is determined by the analytical coupling Equations (16)–(18). ξ and s are, respectively, regarded as additional coordinates relative to the x and y axes. $E^{\prime }$ represents the equivalent physical quantity of MGP and the elastic modulus; Ω denotes the calculation domain at the meshing point of the MGP.

Hybrid TEHL numerical solution problem is determined by the load conditions that have been set; the solution pressure must meet the above-mentioned preconditions for load balance. The load balancing equation is expressed as:(19)$$\iint_{\Omega }p(x,y,t)dxds=F_{T}(t)$$
here, when the parameter h≠0 exists, p denotes the film pressure index of hydrodynamic oil, whereas, when h=0, p is the interface pressure of non-smooth CIMT. For the GTS, $F_{T}$ is the gear interface DMF of a non-double MGP, solved by the above Equation (11).

The friction of the gear meshing interface with TEHL conditions is caused by the hydrodynamic oil film viscous shear stress between the MGP. The aforementioned shear stress is caused by the combination of Poiseuille and Couette flows and varies linearly in real time along the z orientation (that is, along the film thickness orientation of the hydrodynamic oil), and which is denoted as:(20)$$\tau (x,y,z,t)=\frac{\eta^{\#}(x,y,t)}{h(x,y,t)}\left[\mu_{g}(t)-\mu_{p}(t)\right]+\left[z-\frac{1}{2}h(x,y,t)\right]\frac{\partial p(x,y,t)}{\partial x}$$

In view of the fact that the meshing interface is not smooth and under the action of external excitation load, the peak position of the rough interface may have uneven and non-smooth contact, thereby forming a hybrid TEHL state. Therefore, the friction force of GTS consists of two parts. One is that there is a hydrodynamic oil film viscous shear stress between the MGP; the other is that the non-indirect contact of rough peaks leads to the rupture of the oil film on the MGP interface, which weakens the lubrication effect and produces uneven friction thermoelastic behavior. According to the analysis process of hybrid TEHL, the intermediate oil film viscous shear force is regarded as the interface oil film contact friction force; the contact interface real-time transient friction force of the MGP in any time domain can be shown as:(21)$$F_{f}(t)=A\sum_{i}^{m}\sum_{j}^{n}(\mu_{b}p_{cij}(t)+\tau_{ij}(t))$$
where M and N respectively represent the grids number along the x and y directions in the numerical calculation domain. A denotes the grid division zone unit, $\tau_{ij}$ is the interfacial oil film equivalent viscous transient shear force at the middle interface layer (where z=0.5h in the above Equation (20)) on mesh nodes (i,j), $p_{cij}$ is the non-smooth interface contact transient pressure on mesh nodes (i,j), and the non-smooth interface contact friction coefficient $\mu_{b}$ is assumed to be 0.1. Combined Equations (8) and (20), substituted into Equation (21), and the contact transient friction force at meshing interface is derived from the following expression:(22)$$F_{fp}(t)=C_{m}(t)\left(R_{g}(t){\dot{\theta }}_{g}(t)+R_{p}(t){\dot{\theta }}_{p}(t)+{\dot{x}}_{g}(t)-{\dot{x}}_{p}(t)\right)+F_{s}(t)+F_{r}(t)$$
(23)$$F_{fg}(t)=-F_{fp}(t)$$

The CIMT friction coefficient mathematical formula is expressed as:(24)$$\mu_{fp}(t)=\frac{F_{fp}(t)}{F_{T}(t)}$$

Based on the film pressure p and film thickness h (convergent analytical value) of the hybrid TEHL model, the viscous damping $C_{m}(t)$, sliding friction force $F_{s}(t)$ and rolling friction force $F_{r}(t)$ of the MGP are analyzed, which can be expressed as follows:(25)$$\left\{ \begin{array}{l} C_{m}(t)=A\sum_{i}^{m}\sum_{j}^{n}\left[\frac{\eta^{\#}(x,y,t)}{h(x,y,t)}\right]\\ F_{s}(t)=C_{m}(t)\left[{\bar{\mu }}_{g}(t)-{\bar{\mu }}_{p}(t)\right]+A\sum_{i}^{m}\sum_{j}^{n}\left[\mu_{b}p_{cij}(t)\right]\\ F_{r}(t)=A\sum_{i}^{m}\sum_{j}^{n}\left[\frac{1}{2}h(x_{i},y_{j},t)\frac{\partial p(x_{i},y_{j},t)}{\partial x}\right] \end{array}\right.$$
(26)$$\begin{array}{l} J_{p}{\ddot{\theta }}_{p}-\sum_{i=1}^{n}{\left[C_{m}(t)R_{p}(t)\right]}_{i}{\dot{x}}_{p}+\sum_{i=1}^{n}{\left[C_{m}(t)R_{p}(t)\right]}_{i}{\dot{x}}_{g}+\sum_{i=1}^{n}{\left[C_{m}(t)R_{p}^{2}(t)\right]}_{i}{\dot{\theta }}_{p}\\ +\sum_{i=1}^{n}{\left[R_{g}(t)C_{m}(t)R_{p}(t)\right]}_{i}{\dot{\theta }}_{g}=T_{p}-F_{p}(t)r_{p}-\sum_{i=1}^{n}{\left[F_{s}(t)R_{p}(t)+F_{r}(t)R_{p}(t)\right]}_{i} \end{array}$$
(27)$$m_{p}{\ddot{y}}_{p}+c_{py}{\dot{y}}_{p}+k_{py}y_{p}=-F_{p}(t)$$

From the aforementioned Equation (25), it can be revealed that the lubricating oil viscous damping $C_{m}(t)$ is proportional to its equivalent viscosity parameter, but inversely proportional to the hydrodynamic oil film thickness at MGP contact interface. The rolling/sliding friction force $F_{s}(t)$ increases with the increase of the viscous damping $C_{m}$, the relative slip transient velocity $\mu_{g}-\mu_{p}$ of the MTS and the non-smooth concave-convex contact pressure $p_{cij}$. The rolling/sliding friction force $F_{r}(t)$ and the oil film thickness h change in direct proportion to the oil film pressure gradient of the contact interface along the sliding direction of the meshing teeth pair. The friction Equation (23) is calculated from the numerical solution of the hybrid TEHL and substituted into the vibration Equation (8). The frictional dynamic coupling equations of the meshing pair of GTS can be expressed as:(28)$$\begin{array}{l} m_{p}{\ddot{x}}_{p}+\left(c_{px}+\sum_{i=1}^{n}{\left[C_{m}(t)\right]}_{i}\right){\dot{x}}_{p}-\sum_{i=1}^{n}{\left[C_{m}(t)\right]}_{i}{\dot{x}}_{g}+k_{px}x_{p}-\sum_{i=1}^{n}{\left[C_{m}(t)R_{g}(t)\right]}_{i}{\dot{\theta }}_{g}\\ -\sum_{i=1}^{n}{\left[R_{p}(t)C_{m}(t)\right]}_{i}{\dot{\theta }}_{p}=-\sum_{i=1}^{n}{\left[F_{s}(t)+F_{r}(t)\right]}_{i} \end{array}$$
(29)$$\begin{array}{l} J_{g}{\ddot{\theta }}_{g}+\sum_{i=1}^{n}{\left[C_{m}(t)R_{g}^{2}(t)\right]}_{i}{\dot{\theta }}_{g}+\sum_{i=1}^{n}{\left[R_{g}(t)C_{m}(t)R_{p}(t)\right]}_{i}{\dot{\theta }}_{p}+\sum_{i=1}^{n}R_{g}(t){\left[C_{m}(t)\right]}_{i}{\dot{x}}_{g}\\ -\sum_{i=1}^{n}{\left[R_{g}(t)C_{m}(t)\right]}_{i}{\dot{x}}_{p}=T_{g}{}_{g}-\sum_{i=1}^{n}{\left[F_{s}(t)R_{g}(t)+F_{r}(t)R_{g}(t)\right]}_{i}-F_{p}(t)r \end{array}$$
(30)$$m_{g}{\ddot{y}}_{g}+c_{gy}{\dot{y}}_{g}+k_{gy}y_{g}=F_{p}(t)$$
(31)$$\begin{array}{l} m_{g}{\ddot{x}}_{g}+\left(c_{gx}+\sum_{i=1}^{n}{\left[C_{m}(t)\right]}_{i}\right){\dot{x}}_{g}-\sum_{i=1}^{n}{\left[C_{m}(t)\right]}_{i}{\dot{x}}_{p}+k_{gx}x_{g}-\sum_{i=1}^{n}{\left[C_{m}(t)R_{p}(t)\right]}_{i}{\dot{\theta }}_{p}\\ -\sum_{i=1}^{n}{\left[C_{m}(t)R_{g}(t)\right]}_{i}{\dot{\theta }}_{g}=-\sum_{i=1}^{n}{\left[F_{g}(t)+F_{r}(t)\right]}_{i} \end{array}$$

For the above Equations (26)–(31), it can be shown that the rotational and translational DoFs are coupled in Equations (28) and (31) along the OLOA direction. Considering that the gear meshing interface frictional force exists in real time, resulting in the interaction between the translational vibration and the torsional vibration of the GTS along the OLOA orientation, the DMF is described as a function of displacement $y_{i}$ and velocity ${\dot{y}}_{i}$ along the LOA orientation. The six DoFs are coupled in Equations (26)–(29) and are solved jointly by the DMF and frictional forces at the MGP interface. The torsional vibration of GTS and all translational vibrations are in a state of interaction. At each point, the three proposed interfacial micro-texture configurations are appraised. The Reynolds and the thermal interface elastohydrodynamic equations are discretized by the finite element method, and the second-order Lagrangian analysis method is adopted. The microscopic problems of the homogenized model are solved in a homogenized manner at each node of the mesh of the macro-problems. However, this results in a large accuracy of degrees of freedom, and since our goal is to evaluate the accuracy of the model proposed here, we want to eliminate sources of error. Those come from the decoupling of macro equations from micro equations. Nonetheless, this will limit our ability to solve minima problems.

## 4. Simulation and Discussion

In this project, considering the same contact interface contact micro kurtosis and hybrid TEHL conditions, the GTS friction dynamics models of different cases of CIMTs are established. To illustrate this in a more concrete manner, these values are consistent with the parameters of Table 1. The basic structural design parameters of the MGP in these cases are summarized in Table 1; the performance parameters of lubricating oil are reflected in Table 2. Under the experimental conditions of input rated torque 500 Nm and preset speed 1000 r/min, the unknown dynamic characteristics of GTS are studied.

The equivalent stiffnesses of supporting bearings of driving/driven gears in the x and y directions are expressed as $k_{px}=k_{gx}=5.82\times 10^{8}\;N/m$ and $k_{py}=k_{gy}=5.82\times 10^{8}\;N/m$, and the equivalent meshing interface dampings are $c_{px}=c_{gx}=6.53\times 10^{3}\;N/m$ and $c_{py}=c_{gy}=6.53\times 10^{3}\;N/m$, respectively. The TMS of each MGP is solved by the potential energy method; the result is described in Figure 7. Among them, the red line represents the gear meshing stiffness in four meshing alternating cycles, whereas the colored dashed lines with blue and green denote the TMS of gear pair 1 and 2 in any meshing period. What needs to be emphasized here is that the TMS of MGP is the sum of the meshing stiffness of all gear teeth pairs participating in the meshing behavior. The zone where $k_{2}=0$ is named the STC zone. There is gear pair 2 with no meshing in the zone, and the total TMS is significantly smaller than the DTC zone, as gear pair 2 is not in meshing state; the total TMS is not significantly higher than that in the DTC zone.

Substituting the total TMS of the MGP into the dynamic friction coupling equations, the numerical simulation results of the vibration response of the GTS is derived. On this basis, the DMFs of the MGPs are determined by Equations (10) and (11). The DMFs are reflected in Figure 8. The pink solid line is regarded as the total DMF of the MGP, whereas the dashed lines with green and blue represent the DMF of the MGPs 1 and 2, respectively. What is of concern here is that the DMF of the MGP expresses the sum of the DMFs of all meshing gear teeth. Furthermore, the following law of change can be inferred: the fluctuation range of the total DMF shows a trend of first decreasing and then increasing. One of the main reasons is that the meshing transient process is caused by the DTC zone crossing into the STC zone.

### 4.1. Friction and Damping between CIMTs under Cases

The vibration response is solved based on the gear dynamics model to determine the transient DMF, the rolling/sliding relative velocity of the gear teeth interface and the curvature radius at any meshing position. The aforementioned values are included in the hybrid TEHL model along with the microscopic morphology and lubricating oil characteristics at the MGP interface to evaluate the friction response, thickness and pressure distribution of the lubricating oil film of the GTS. For the sake of comparison, a specific case of a smooth gear tooth interface is provided, as shown in Figure 9 and Figure 10.

Figure 9 reveals the influence of the law on the CIMT geometric parameters and configuration distribution characteristics on the viscous damping index, as the parameter characterization wavelength L is 20 μm and the MTS contact micro kurtosis $R_{q}$ value is 1.0 μm. The solid lines with red, blue, and green separately represent the damping index consistent with CIMT cases 1, 2 and 3, whereas the solid line with black denotes a smooth MTS.

Table 3 reveals the viscous damping values at the five specified points (A, B, C, D and E represent the initial starting point, STC to DTC the first half transition point and the pitch point, DTC to STC the second half transition point, and the existing point) under the three conditions mentioned above. From Case 1 to Case 3, a significant finding is that, as CIMT becomes more regular and orderly, the damping value becomes higher. The reason is that, compared with Case 1, the MTS contact micro kurtosis peaks in Case 3 are more evenly concentrated near the contact centerline, which leads to a closer connection between the MGP contact interfaces. Therefore, the oil film thickness of Case 3 is thinner than that of Case 1. Furthermore, the smooth surface produces a higher viscous damping effect than the three rough MTS cases, which can be attributed to the thinnest hydrodynamic oil film produced between the no micro-convex topography MTSs. Another noteworthy finding is that the meshing interface damping value of the DTC zone is not lower than that of the STC zone. This reason is attributed to the fact that the slip speed between the gear meshing interfaces of the DTC zone is not higher than that of the STC zone, which makes it easier to form an oil film. Figure 10 depicts the influence of the CIMT geometric distribution as the wavelength L value is 20 μm and the MTS contact micro kurtosis $R_{q}$ value of the MGP is 1.0 μm. The solid lines with red, blue, and green, respectively, describe the transient friction consistent with CIMT cases 1, 2 and 3, whereas the solid line with black represents a smooth MTS. It can be revealed that for the three non-smooth MTS cases, the transient friction force of the DTC zone is less than that of the STC zone. As the CIMT geometric parameters and configuration distribution characteristics cases 1, 2 and 3 gradually slowed down, the peak of the MTS contact micro kurtosis decreased, which leads to the gradual weakening of the rough contact with the micro-textured interface. Therefore, the corresponding changes of friction force gradually show a decreasing trend from case 1 to case 3. Whether it is in the STC or DTC zone, the friction generated by the smooth meshing gear tooth surface is the lowest, which is the “smooth” contact that is incorporated into the MTSs. Especially at the nodes, the friction between smooth MTSs is regarded as zero. The following reveals the correlation effect of the wavelength L valve (with a fixed $R_{q}$) under the three cases of CIMT on the transient friction and time-varying viscous damping of the MTS. Figure 11(a_1_–a_3_) reflects the friction trend of the three CIMT cases 1, 2 and 3 described in Figure 12 at multiplex wavelengths L, where Figure 11(b_1_–b_3_) reveals the CIMT viscosity viscous damping absorption effect.

The friction and damping of CIMT in the three cases were averaged within a meshing period, the correlation law of the CIMT case and wavelength on average transient friction force was analyzed, and the relative influence of average transient friction force and time-varying damping was explored. The average friction trend of different types of CIMT within a certain wavelength range follows Figure 12a. It can be seen from the above figures that, as the wavelength is slight, the difference in average friction consistent with CIMT case 2 and case 3 is extremely small. As the wavelength becomes larger, the average friction of CIMT case 2 is not less than that of case 3. As the wavelength increases, the friction force (average value) of the CIMT case 1 drops slightly at a certain point (wavelength is about 30 μm), and then, due to the randomness of CIMT in case 1, it fluctuates sharply nearby. The average friction force of CIMT Cases 2 and 3 suddenly decreased to a minimum (wavelength is approximately 14 μm) and followed the law of increasing with the increase of the wavelength. In turn, this can verify the key role of CIMT wavelength on the friction effect between MTSs. Figure 12b illustrates the changes of viscous damping (average value) under CIMT cases in multiplex wavelength ranges; an average viscous damping is not greatly affected by the CIMT wavelength valve. The viscous damping (average value) consistent with CIMT Case 2 is approximately the same as that of Case 3, but not less than that of Case 1.

### 4.2. CIMT Dynamic Characteristics under Cases

The proposed model considers the mutual influence of CIMT under different cases on the dynamic characteristics of MTS. Figure 13 reveals the correlation of the displacement response of driving gear MTS with CIMT wavelength valve (L = 20 μm) along the OLOA orientation relative to the contact micro kurtosis value (RMS) and input design torque in the three cases. It can be concluded that the contact micro kurtosis value (RMS) decreases from CIMT case 1 to case 3, and the decrement increases as the torque increases. It was further learned that the contact micro kurtosis value (RMS) difference between any regular MTS (i.e., Cases 2 or 3) and random permutation MTS (i.e., Case 1) is not less than the contact micro kurtosis value (RMS) difference between two regular MTSs (i.e., Cases 2 and 3) and that the contact micro kurtosis value (RMS) has relatively little influence on the dynamic interface contact response.

## 5. Conclusions

TMS, STAE, and CIMT geometric distribution and oil film shear effects were taken into consideration. According to the dynamic control equation of the six-DoFs MGP, the friction dynamic coupling model of the meshing interface of the MGP and the 3D TEHL contact model were established. The friction dynamics coupling model was solved by iterative loop until it converged. The coupling model proposed above was implemented to study the geometric distribution influence on MGP with CIMT of the dynamic characteristics under MTS TEHL conditions. From this, the following conclusions were inferred:
(1)As the CIMT changes from random permutation (Case 1) to regular (Artificial routines, Cases 2 and 3), the MTS viscous damping becomes higher, and the not-rough MTS has the highest damping value. Conversely, the transient frictional force decreases with the smoothness of the MTS.(2)For a regular MGP, the CIMT wavelength quantification characterization plays a vital role in the i interfacial frictional transient contact effect between MTS, and the frictional force drops to an extreme value at a specific wavelength (the study case is about 14 μm).(3)The MTS dynamic response along the OLOA direction decreases as CIMT becomes regular (that is, from Case 1 to Case 3), which means that the production of a regular shape (rather than a random shape) meshing gear tooth surface is more suitable for reducing the vibration response of the GTSs.


The next focus of future work is to conduct experimental research on the dynamic behavior of gear MTSs with multiplex types of CIMT, to deepen the verification of relevant theoretical findings through the research of this sub-project.

## Figures and Tables

**Figure 1 materials-15-02075-f001:**
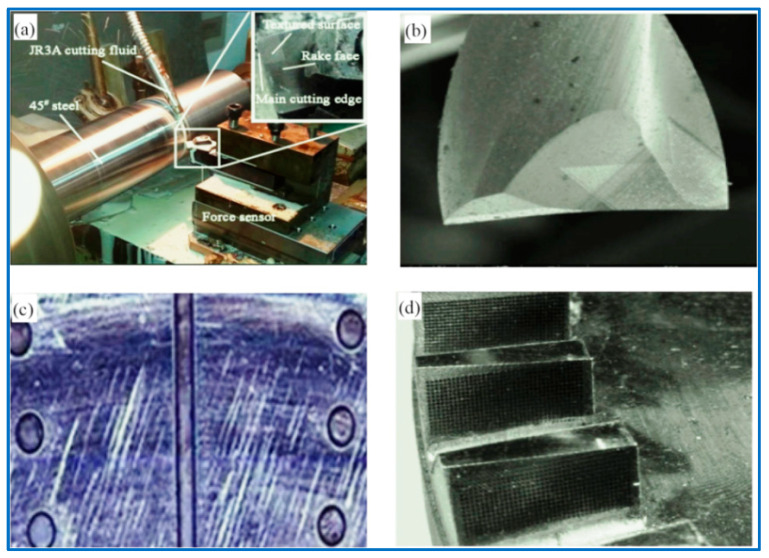
CIMT representations (**a**,**b**) A tool surface, (**c**) Cylinder inner liner surface, (**d**) MGP surface.

**Figure 2 materials-15-02075-f002:**
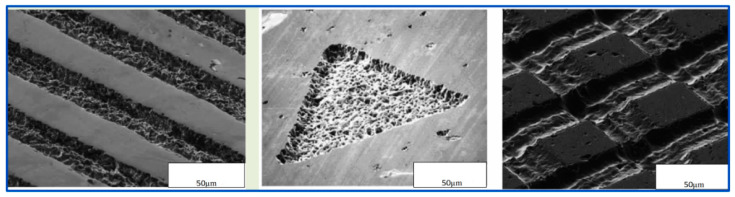
Bionic micro-textured surface.

**Figure 3 materials-15-02075-f003:**
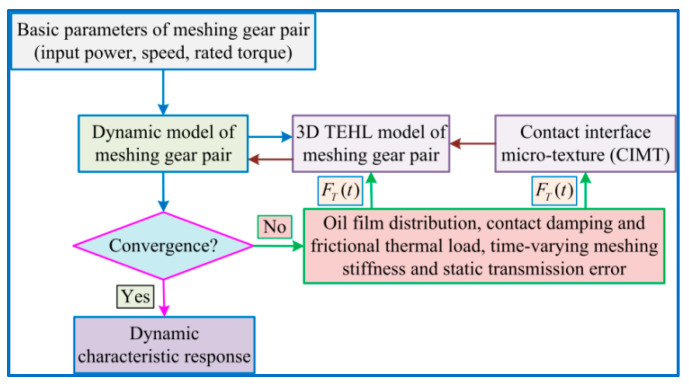
Flowchart of CIMT calculation method.

**Figure 4 materials-15-02075-f004:**
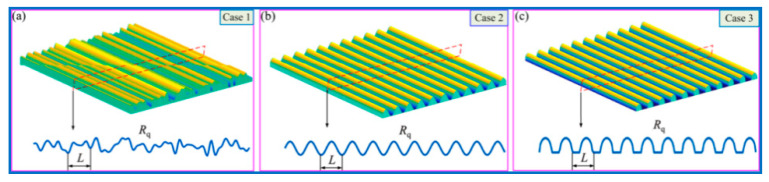
Three cases of CIMT configuration of MTS. (**a**) Case 1, (**b**) Case 2, (**c**) Case 3.

**Figure 5 materials-15-02075-f005:**
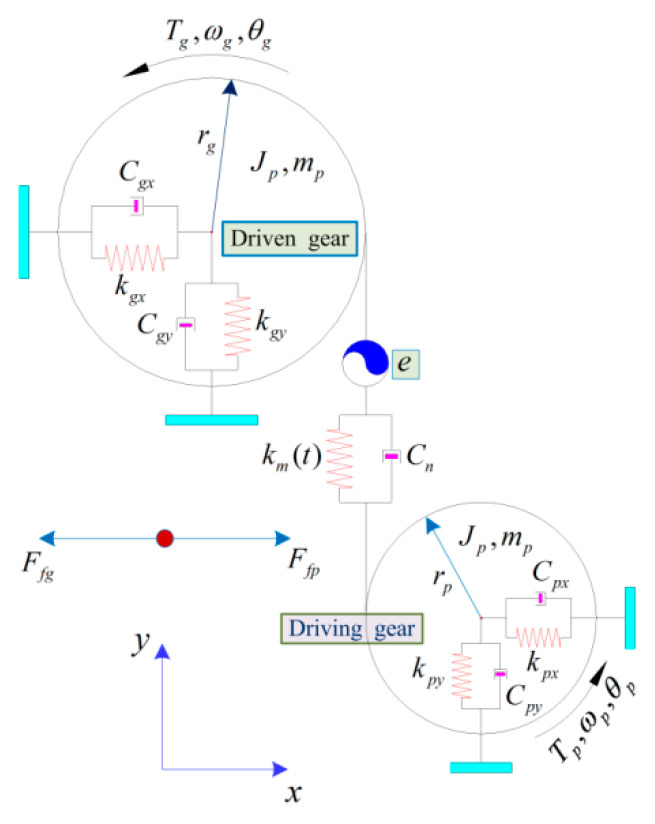
A general spur gear dynamic model considering MTS friction behavior.

**Figure 6 materials-15-02075-f006:**
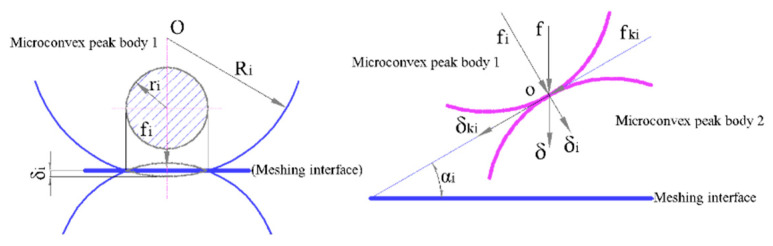
Schematic representation of geometric distribution of MGPs composite CIMT at the meshing point.

**Figure 7 materials-15-02075-f007:**
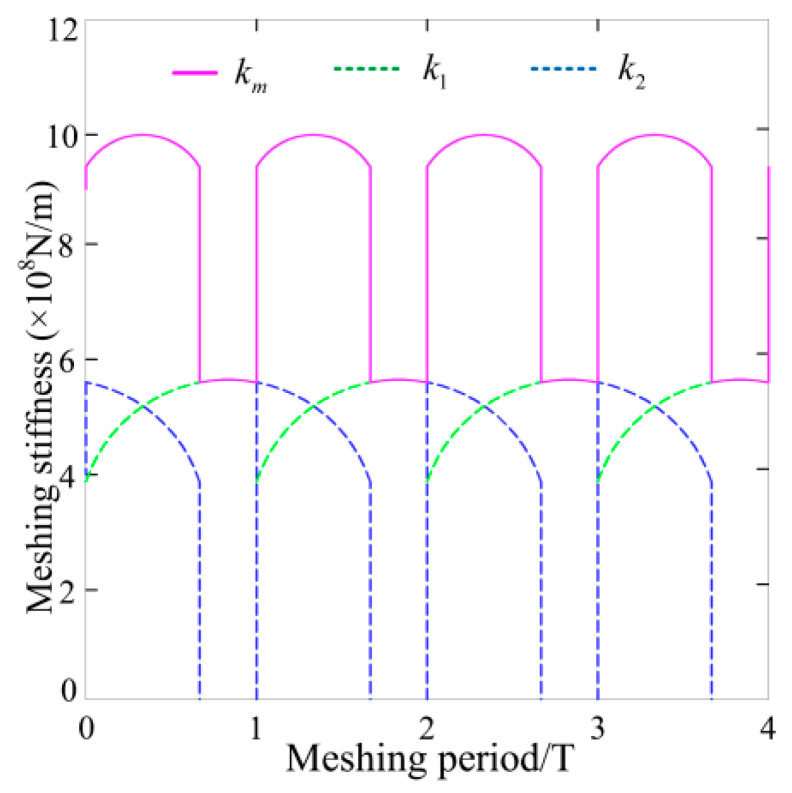
Each MGP and its TMS curves.

**Figure 8 materials-15-02075-f008:**
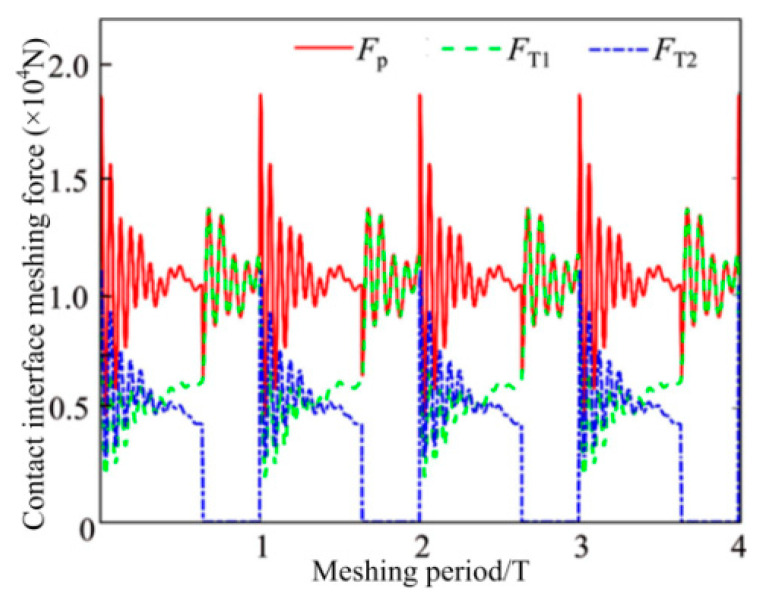
Dynamic gear tooth DMF of each MGP.

**Figure 9 materials-15-02075-f009:**
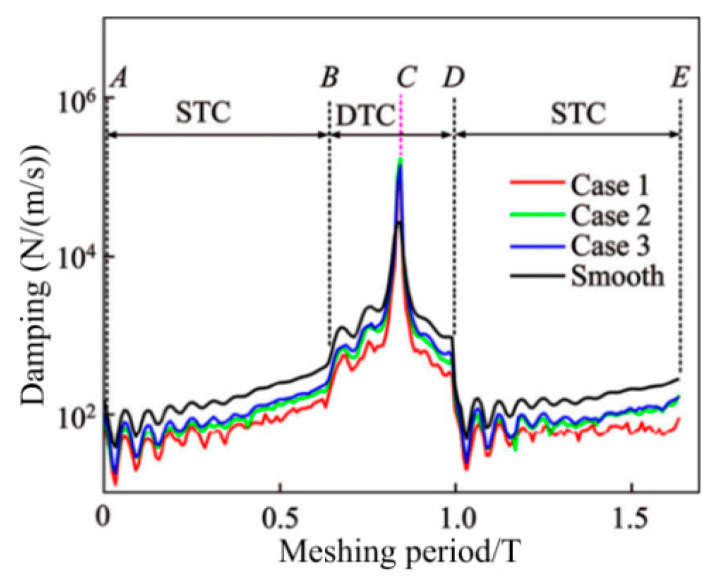
Viscous damping of MGPs with CIMT.

**Figure 10 materials-15-02075-f010:**
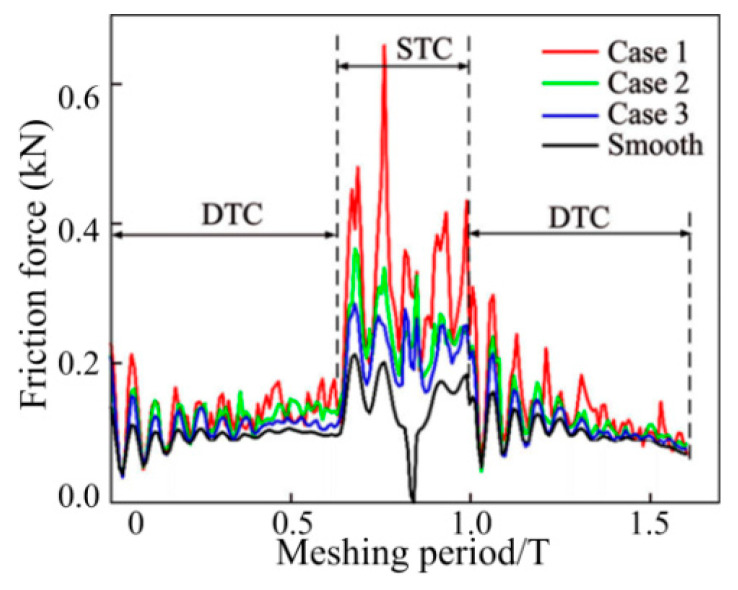
Friction forces of MGPs with CIMT.

**Figure 11 materials-15-02075-f011:**
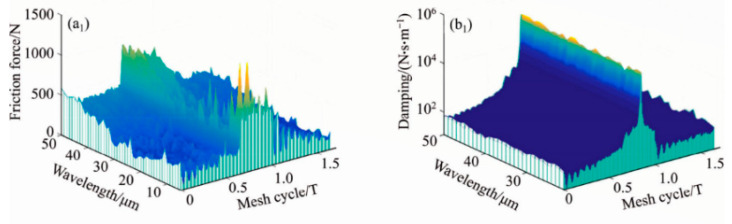
Transient friction force and time-varying viscous damping with CIMT cases of MGPs of multiplex wavelengths (**a_1_**,**b_1_**) Case 1, (**a_2_**,**b_2_**) Case 2, (**a_3_**,**b_3_**) Case 3.

**Figure 12 materials-15-02075-f012:**
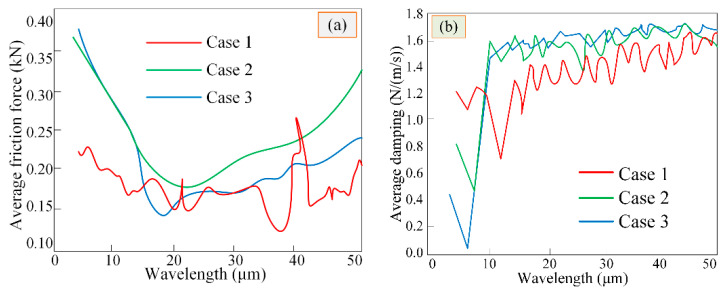
Averaged values with multiplex wavelengths (**a**) Frictional forces, (**b**) Damping values.

**Figure 13 materials-15-02075-f013:**
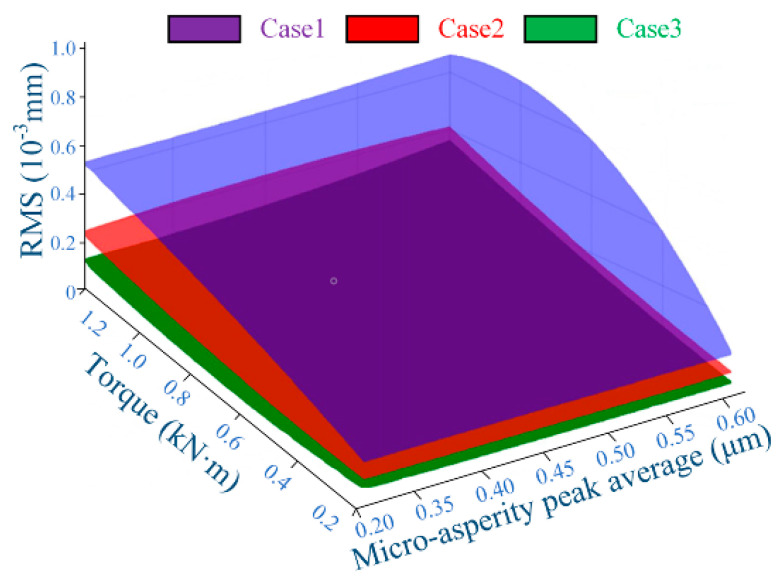
Displacement dynamic response along OLOA MTS with CIMT contact micro kurtosis R_q_ and RMS of input torque.

**Table 1 materials-15-02075-t001:** Basic design parameters of MGP.

Parameter	Value
Number of gear teeth, z_p_/z_g_	27/33
Module, M/mm	4.5
Center distance, d_c_/mm	116
Pressure angle, α/(°)	20
Face width, w/mm	45
Clearance coefficient, c	0.25
Addendum coefficient, h_a_	1.0
Meshing interface R_q_ value/μm	0.5
Elastic modulus, E/GPa	207
Density, (ρ_p_/ρ_g_)/(10^3^ kg·m^−3^)	7.85/7.85
Poisson ratio, μ	0.31
Mass, (m_p_/m_g_)/kg	2.89/3.02
Moment of inertia, (J_p_/J_g_)/ (10^−3^ kg·m^3^)	5.4/7.8
STAE, e/μm	1.0

**Table 2 materials-15-02075-t002:** Performance parameters of lubricating oil.

Parameter	Value
Reference shear stress, τ_0_/MPa	18.2
Inlet viscosity, η_0_/(Pa·s)	0.10
Viscosity-pressure index, z_0_	0.71
Inlet density, ρ_0_/(kg·m^−3^)	886

**Table 3 materials-15-02075-t003:** Viscous damping values of 5 specified points in a meshing period under cases.

Cases	Damping/(N·s·m^−1^)				
	A	B	C	D	E
1	51.7	172.9	1.092 × 10^3^	113.9	91.03
2	61.6	231.2	1.663 × 10^3^	154.2	165.1
3	75.7	281.6	1.339 × 10^3^	194.8	174.2

## Data Availability

Not applicable.

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
