# Peer review of "Research on Meshing Gears CIMT Design and Anti-Thermoelastic Scuffing Load-Bearing Characteristics"

_materials, 2022, doi:10.3390/ma15062075_

Round 1
Reviewer 1 Report
The article deals with the modeling of the friction wear mechanism of the meshing gears contact, taking into account microtexture of interface. The authors declared aim was to create two models: contact characteristics and coupled contact friction dynamics.
In my opinion, the introduction is very comprehensive and sufficient. The next two chapters describe gear dynamic model with CIMT and 3D TEHL model. The fourth chapter contains simulated object parameters and discussion of results. The conclusions are broad and clear.
The bibliography consists of 35 items - all fresh: 5 item from the last year, 29 from the previous five years and 1 item after 2000.
General remark:
Both models mentioned in the article are abundantly described with both kinematic diagrams and impressive systems of differential equations, but the formal part concerning simulation contains only the simulation output parameters, visualized results and discussion, but there is no information on the methods of solving equation systems or also determining the uncertainty of solutions. I strongly suggest supplementing this information, because the present form of presentation does not justify including this extensive theoretical part - there is a gap between it and the results.
Minor remarks:
1. All graphs should be in vector format as the current raster form is of poor quality.
2. I suggest improving the style of some sentences as they are difficult to understand.
3. Line 563 - remove template phrase "Please add:"
Author Response
Dear reviewer,
Thank you very much for your comments and suggestions, these comments are fair, encouraging, and constructive. My team members and I have learned much from it. We submit here the revised manuscript and our responses to your comments. Please refer to the attachment. We sincerely hope that we will have the opportunity to publish our works.
Best wishes,

Reviewer 2 Report
The manuscript was written with an interesting topic in a scientifically relevant field. However, all abbreviations must begin with a capital letter when are first mentioned in the manuscript such as: "contact interface micro-texture" must be "Contact Interface Micro-Texture (CIMT)", "gear transmission system (GTS)" must be "Gear Transmission System (GTS)", "static transmission error (STE)" must be "Static Transmission Error (STE)", etc..
Author Response
Comments and Suggestions for Authors:
The manuscript was written with an interesting topic in a scientifically relevant field. However, all abbreviations must begin with a capital letter when are first mentioned in the manuscript such as: "contact interface micro-texture" must be "Contact Interface Micro-Texture (CIMT)", "gear transmission system (GTS)" must be "Gear Transmission System (GTS)”, “static transmission error (STE)" must be "Static Transmission Error (STE)", etc.
Sincerely respond to the above suggestions raised by reviewer 2:
The authors are very grateful to the Reviewer 2 for their reasonable suggestions and valuable comments on this manuscript. According to the questions raised by the reviewer, the authors have conducted a comprehensive and detailed inspection and correction of the full text of the manuscript, especially the grammatical errors and poor language polishing of this manuscript. Through this revision of the manuscript, the authors have obtained many improvements and advancements in the process, and sincerely hope that the English writing level and language quality of the manuscript submitted in the future will gradually approach the requirements and standards of your journal.
Reviewer 3 Report
The authors have prepared a good manuscript. I am happy to recommend the publication without any revision. However, it will be good if the manuscript can be re-checked for any existing language related minor issues.
Author Response
Comments and Suggestions for Authors
The authors have prepared a good manuscript. I am happy to recommend the publication without any revision. However, it will be good if the manuscript can be re-checked for any existing language-related minor issues.
Sincerely respond to the above suggestions raised by reviewer 3:
The authors are very grateful to the Reviewer 3 for their reasonable suggestions and valuable comments on this manuscript. According to the questions raised by the reviewer, the authors have conducted a comprehensive and detailed inspection and correction of the full text of the manuscript, especially the grammatical errors and poor language polishing of this manuscript. Through this revision of the manuscript, the authors have obtained many improvements and advancements in the process, and sincerely hope that the English writing level and language quality of the manuscript submitted in the future will gradually approach the requirements and standards of your journal.
Reviewer 4 Report
The paper presents a very complex dynamic mathematical model. However, it is not considering some important aspects like gear accuracy (pitch tolerances, tooth line accuracy etc.). You should mention this from the beginning.
There are some other recommendations as minor revisions
- Lines 217-222 Case 1, 2 and 3 are actually the CIMT. It is the topology of one surface only if the other surface in contact is smooth.
- The sentence from line 364 is a title? Looks like it is not finished.
- Figure 9 – STC and DTC regions are placed incorrectly
- Figure 13 and discussion on Figure 13 are ambiguous: You are using the terms Rq and RMS, like they are something different. You should use along the entire paper only Rq or only RMS. What is RNS on Figure 13? What is Roughness on Figure 13.
For the measurement units in all the diagrams, you are using two forms, with or without brackets (μm) or μm. Keep it unitary.
